# Relationships of global longitudinal strain with s', long-axis systolic excursion, left ventricular length and heart rate

**Roger E. Peverill** *, **Kevin Cheng** , **James Cameron, Lesley Donelan, Philip M. Mottram**

Monash Cardiovascular Research Centre, MonashHeart and Department of Medicine (School of Clinical Sciences at Monash Health), Monash University and Monash Health, Clayton, Victoria, Australia

* roger.peverill@monash.edu

## Abstract

### Background

Longitudinal left ventricular (LV) contraction can be impaired in the presence of a normal LV ejection fraction (LVEF), and abnormalities have been reported in global longitudinal strain (GLS), long-axis systolic excursion (SExc), and the peak systolic velocity (s') of mitral annular motion using tissue Doppler imaging (TDI). However, the relationships of GLS with s' and SExc have not been systematically evaluated in subjects with a normal LVEF, and whether these relationships might be affected by variations in LV end-diastolic length (LVEDL) and heart rate is unknown.

### Methods

We investigated the univariate and multivariate correlations of GLS with TDI measurements of s' and SExc (both using averages of the septal and lateral walls), LVEDL and heart rate in subjects with a normal LVEF (>50%) but a low peak early diastolic mitral annular velocity (septal e'$\leq$ 7.0 cm/s and lateral e'$\leq$ 9 cm/s), and thus an increased risk of a future cardiac event.

### Results

84 subjects (age 66±8 years, 29 males) with a LVEF of 62±6% and GLS of -17.5±2.3% were studied. On univariate analysis the absolute value of GLS was positively correlated with s'(r = 0.28, p<0.01) and SExc (r = 0.50, p<0.001) and inversely correlated with heart rate (r = -0.36, p = 0.001), but was not correlated with LVEDL (r = -0.15). In multivariate models, SExc explained more of the variance in GLS than s', and absolute GLS was not only positively correlated with SExc, but also inversely correlated with LVEDL. Heart rate was an independent inverse correlate of GLS in conjunction with LVEDL and either s' or SExc, but made a larger contribution in models which included s'. Interobserver correlations were close for s' and SExc (r = 0.89–0.93), but only moderate for GLS (r = 0.71).

**Funding:** The authors received no specific funding for this work.

**Competing interests:** The authors have declared that no competing interests exist.

**Abbreviations:** BP, blood pressure; e', peak velocity of early diastolic mitral annular motion; GLS, global longitudinal strain; HFpEF, heart failure with preserved ejection fraction; LV, left ventricular; LVEDL, left ventricular end-diastolic length; LVEF, left ventricular ejection fraction; s', peak velocity of systolic mitral annular motion; SExc, systolic mitral annular excursion; SS, systolic signal; TDI, tissue Doppler imaging.

## Conclusion

In subjects with a normal LVEF but reduced e', the absolute value of GLS is more closely related to SExc than s', and is also independently and inversely related to LVEDL and heart rate. Measurement of SExc may provide a useful additional or alternative technique to GLS for the assessment of LV long-axis function.

## Introduction

Left ventricular (LV) long-axis contraction can be impaired in the presence of a normal LV ejection fraction (LVEF), and this has been demonstrated using a number of echocardiographic techniques including M-mode [1–3], tissue Doppler imaging (TDI) [4–6] and both colour Doppler and speckle tracking strain [3, 7–11]. That there is clinical significance of this impairment is suggested by evidence that reduced LV long-axis contraction is one of the features of heart failure with preserved EF (HFpEF) [3, 8, 9, 11], and supported by reports that there is prognostic utility for measurements of mitral annular systolic excursion (SExc) [12, 13], the peak mitral annular systolic velocity (s') [14, 15], and global longitudinal strain (GLS) [11, 16, 17] in subjects with a normal LVEF. A deterioration in LV long-axis systolic function is also recognized to be an early indicator of a subsequent reduction of LVEF in cardio-oncology, and there has been particular interest in the use of GLS measurements using speckle tracking for this purpose [18]. Suggested advantages of GLS over M-mode and TDI variables include its relative independence from translation, tethering and the angle of incidence. However, there are also limitations of GLS, with a major limitation being that accurate measurement is dependent on having non-foreshortened LV imaging in all three apical LV views, in conjunction with imaging of adequate quality to allow tracking in most of the 6 segments in each of these views. Indeed, in large studies in subjects with normal LVEF, GLS has not been feasible in a substantial proportion of the subjects [9, 11, 17], with the implication that it cannot be used for the assessment of LV long-axis systolic function in all individuals. The lack of general feasibility of GLS is in contrast with mitral annular TDI signals, for which measurements have been possible in most subjects in large population studies [15, 19].

While there must be overlap in the information about LV long-axis function provided by systolic TDI variables and GLS, a detailed understanding of the relationships between the different long-axis variables is necessary prior to determination of their interchangeability and relative utilities. However, there has only been limited study of the relationships between GLS, SExc and s' in the setting of a normal LVEF. There is one report of a positive correlation of the absolute value of GLS (conventionally a negative number) with M-mode derived SExc in an analysis of HFpEF and control subjects [3], two reports of positive correlations of absolute GLS with s' in subjects at their baseline assessment in HFpEF trials [9, 11], and a recent population study reported reductions in both absolute GLS and M-mode derived SExc during aging [20]. A limitation of these studies has been that the relationships of GLS with SExc and s' have not been compared within the same study, this being of potential importance given that the correlation of s' with SExc has been variable (r = 0.47–0.71) in previous studies [3, 21]. Neither has there been consideration in most studies of whether variations in left ventricular end-diastolic length (LVEDL) and heart rate might affect the relationship of GLS with s' or SExc. The aim of the present study was to investigate the relationships of GLS with TDI measurements of SExc and s', LVEDL and heart rate in a group of subjects without a history of heart failure, but considered to be at risk of future cardiac events due to the presence of a reduced LV mitral

annular peak early diastolic velocity (e') [15, 22, 23]. Adults with low values of septal and lateral e'were identified from a group of subjects who had been referred for the investigation of chest pain, dyspnea or possible ischemic heart disease, and found to have a normal LVEF, normal short-axis regional LV contraction, no evidence of ischemic heart disease, no significant valvular disease, and LV 2D imaging which appeared to be of suitable quality for the measurement of GLS.

## Methods

### Subjects

The study design was approved by the Monash Health Human Research Ethics Committee and all clinical investigation was conducted according to the principles expressed in the Declaration of Helsinki. Studies were identified retrospectively from consecutive outpatient stress echocardiograms performed at Monash Health for clinical indications between July 2011 and May 2013. The need for individual patient consent was waived. Exclusion criteria were known ischemic heart disease, a LVEF <50%, a regional LV wall motion abnormality, a cardiac rhythm other than sinus, an atrioventricular conduction abnormality or bundle branch block, evidence of inducible myocardial ischemia, more than mild valvular dysfunction and inadequate image quality for the identification of endocardium or epicardium in any of the three apical views of the left ventricle. There were 87 studies identified in which the exclusion criteria were absent and in which there was a low e', based on a septal e'≤ 7.0 cm/s and a lateral e'≤ 9.0 cm/s, of which 3 were subsequently excluded because of the presence of more than 2 segments in any of the 3 views in which strain could not be measured.

### Echocardiography

Echocardiography was performed prior to exercise with the subject in a supine position using a GE Vivid 7 (GE Healthcare, Chicago, Illinois, USA). Apical 4-, 2- and 3-chamber two-dimensional loops of LV contraction were recorded. Pulsed wave tissue Doppler imaging was performed at the septal and lateral borders of the mitral annulus. All the GLS and TDI measurements were performed off-line, with GLS and systolic TDI measurements made at separate sittings by independent investigators who were blinded to GLS results when making TDI measurements. GLS was measured as peak-systolic strain using 2D speckle tracking in the apical 4-, 2- and 3-chamber views (EchoPac V 113.1.5, GE Healthcare, Chicago, IL, USA) consistent with previously described principles [24]. The LVEF was calculated using the biplane method of discs. The LV length at end-diastole from the plane of the mitral annulus to the apical endocardium in the 4- and 2-chamber views was recorded during the measurement of the LV end-diastolic volume, and the longest dimension from these 2 views has been used as the LVEDL [25, 26]. Measurements of the TDI signals were made of e', s', and of SExc as the integral of the systolic signal during ejection, with the tracing performed just inside the outer border of the Doppler envelope, as previously described [26, 27].

**Measurement variability.** All TDI values shown represent an average of the measurements from the septal and lateral annular borders. Averages of the GLS and TDI results from two independent investigators have been used in the regression analyses investigating the relationships between GLS and TDI variables. Variability between the two observers has been assessed for the TDI and GLS measurements using linear regression analysis, the mean differences and Bland-Altman plots.

## Statistical analysis

Statistical analysis was performed using Systat V13 (Systat Software, Chicago, IL, USA). Continuous variables are presented as mean ± SD. Univariate linear regression analysis was performed to determine the correlates of GLS (using the absolute value of GLS) and the r value has been reported. Multivariate regression analysis was performed with variables chosen and entered sequentially to address the aims of the study. The partial correlation coefficient (β) value is provided for variables in the multivariate analyses. The coefficient of determination has been adjusted for the number of terms in the model (adjusted $r^2$) and used to estimate the extent of variance in a dependent variable explained by a multivariate model. A *p* value of <0.05 has been accepted as significant.

## Results

Demographic, anthropometric and echocardiographic data of the study group are shown in Table 1. The LVEF varied between 50.7% and 76.3% and GLS varied between -12.7% and -24.6%. Univariate correlations of GLS are shown in Table 2. The absolute value of GLS was positively correlated with s'(Fig 1) and SExc (Fig 2) and inversely correlated with heart rate, a positive correlation of GLS with systolic BP was borderline significant, but GLS was not correlated with LVEDL.

Multivariate models of the absolute value of GLS which include combinations of s', SExc and LVEDL are shown in Table 3. In the multivariate model which included both s' and SExc, a greater proportion of the variance in GLS was explained by SExc than by s', and s' was no longer a significant contributor to the model. In the model which included LVEDL with SExc, SExc was a positive correlate of GLS, LVEDL was an inverse correlate, and there were increases in the partial correlation coefficients for both SExc and LVEDL when these variables were

**Table 1. Subject characteristics.**

| | |
|---|---|
| Male: Female | 29:55 |
| Age (years) | 66±8 |
| Hypertension | 61 (73%) |
| Diabetes | 23 (27%) |
| Indication for stress echocardiogram | |
| Chest pain | 52 (62%) |
| Dyspnea | 9 (11%) |
| Other | 23 (27%) |
| Body surface area (m$^2$) | 1.81±0.21 |
| Body mass index (kg/m$^2$) | 28.9±4.2 |
| Blood pressure (mmHg) | 137±18/80±10 |
| Heart rate (bpm) | 73±9 |
| Left atrial volume index (mL/m$^2$) | 33±7 |
| LVEF (%) | 62±6 |
| GLS (%) | -17.5±2.3 |
| LVEDL (cm) | 8.5±0.7 |
| Transmitral E/A | 0.88±0.23 |
| Average s'(cm/s) | 6.7±1.2 |
| Average SExc (cm) | 1.2±0.2 |
| Average e'(cm/s) | 5.2±0.9 |

**Table 2. Univariate correlations with the absolute value of GLS.**

| Independent variable | r | p |
|---|---|---|
| SExc | 0.50 | <0.001 |
| s' | 0.28 | 0.011 |
| LVEDL | -0.15 | 0.18 |
| Heart rate | -0.36 | 0.001 |
| Systolic blood pressure | 0.21 | 0.054 |
| Diastolic blood pressure | 0.12 | 0.29 |

combined. In the model of GLS which included s' and LVEDL, LVEDL became a borderline significant contributor.

To explore possible reasons for the closer correlation of GLS with SExc than with s', and in view of previous evidence that heart rate has a modifying effect on the relationship between s' and SExc [21], the relationships of s' with SExc, and of s' and SExc with heart rate, were examined. On univariate analysis, there was only a moderate correlation of s' with SExc (r = 0.50, p<0.001) and heart rate was inversely correlated with SExc (r = -0.41, p<0.001), but not correlated with s'(r = 0.17, p = 0.12). In a multivariate model of s', after the addition of heart rate to SExc, SExc remained a positive correlate (β = 0.68, p<0.001), heart rate also became a positive correlate (β = 0.45, p<0.001), and there were increases in the partial correlation coefficients of both independent variables. Moreover, the variance of s' explained following the addition of heart rate to SExc increased substantially from to 24% to 40%.

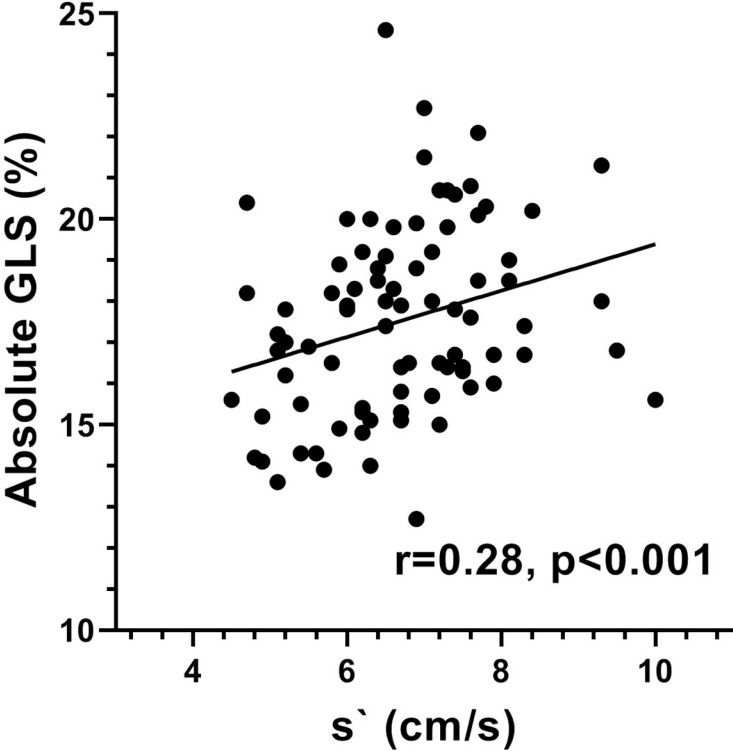

**Fig 1. Scatter plot showing the relationship of the absolute value of GLS with s'.**

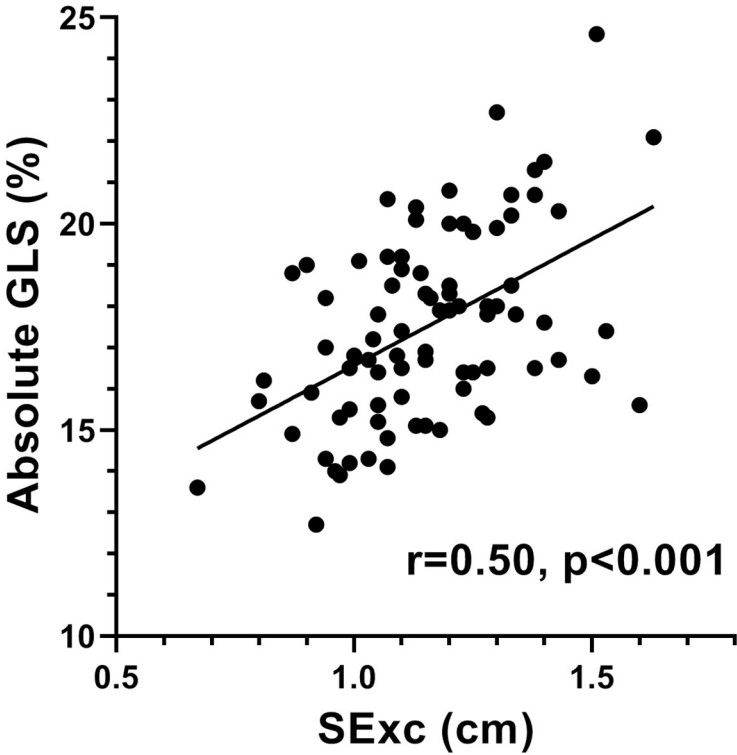

**Fig 2. Scatter plot showing the relationship of the absolute value of GLS with SExc.**

Multivariate models of GLS which include combinations of heart rate and systolic BP with s', SExc and LVEDL are shown in Table 4. In combination, heart rate and systolic BP were independent predictors of GLS and there was no interaction evident between these two variables when they were combined. The addition of heart rate to s' and LVEDL resulted in a substantial increase in the variance of GLS explained, and in turn, the addition of systolic BP provided a further mild increase, with all variables significant contributors in the model. In contrast, although heart rate was also a significant contributor to the prediction of GLS in combination with SExc and LVEDL, its inclusion only resulted in a small increase in the variance of GLS explained, and systolic BP was not a contributor to the prediction of GLS in combination with SExc, LVEDL and heart rate. The model which included s', LVEDL, heart rate and systolic BP explained the same variance of GLS (38%) as the model with SExc, LVEDL and heart rate.

Results of GLS, s' and SExc from the two observers showing means, correlations and differences are shown in Table 5 and Bland-Altman plots are shown in Fig 3. There were small

**Table 3. Models of the absolute value of GLS including s', SExc and LVEDL.**

| Independent variable | r | β in multivariate model | p value in multivariate model | Cumulative adjusted $r^2$ |
|---|---|---|---|---|
| s' | 0.28 | 0.044 | 0.69 | 0.07 |
| SExc | 0.50 | 0.47 | <0.001 | 0.23 |
| s' | 0.28 | 0.31 | 0.005 | 0.07 |
| LVEDL | -0.15 | -0.19 | 0.08 | 0.09 |
| SExc | 0.50 | 0.59 | <0.001 | 0.24 |
| LVEDL | -0.15 | -0.33 | 0.001 | 0.33 |

**Table 4. Models of the absolute value of GLS including heart rate and blood pressure.**

| Independent variable | r | β in multivariate model | p value in multivariate model | Cumulative adjusted $r^2$ |
|---|---|---|---|---|
| Heart rate | -0.36 | -0.37 | <0.001 | 0.12 |
| Systolic BP | 0.21 | 0.23 | 0.026 | 0.16 |
| s' | 0.28 | 0.35 | 0.001 | 0.07 |
| Heart rate | -0.36 | -0.42 | <0.001 | 0.23 |
| s' | 0.28 | 0.42 | <0.001 | 0.07 |
| Heart rate | -0.36 | -0.54 | <0.001 | 0.24 |
| LVEDL | -0.15 | -0.36 | <0.001 | 0.34 |
| s' | 0.28 | 0.42 | <0.001 | 0.07 |
| LVEDL | -0.15 | -0.35 | <0.001 | 0.09 |
| Heart rate | -0.36 | -0.55 | <0.001 | 0.34 |
| Systolic BP | 0.21 | 0.22 | 0.013 | 0.38 |
| SExc | 0.50 | 0.50 | <0.001 | 0.21 |
| LVEDL | -0.15 | -0.37 | <0.001 | 0.33 |
| Heart rate | -0.36 | -0.26 | 0.008 | 0.38 |

differences in the average values of GLS, s' and SExc between the observers. Interobserver correlations were close for s' and SExc (r = 0.89–0.93), but only moderate for GLS (r = 0.71).

An average of the apical 4-chamber GLS from the 2 observers was also investigated on the basis that 4 chamber view derived TDI variables might be more closely correlated with GLS from the 4-chamber view than from standard GLS which is an average from the 3 apical views. There was a small systematic difference between GLS and 4-chamber GLS (-16.9% v -17.5% p<0.001), but the two variables were closely correlated (r = 0.85, p<0.001). The closeness of the relationships of GLS with SExc were examined comparing apical 4-chamber GLS and standard 3-view GLS, with LVEDL and heart rate included in the models. Only 30% of the variance in SExc was explained by the model which included 4-chamber GLS, whereas the addition of standard GLS to this model improved the variance of SExc explained to 40%.

## Discussion

The aim of this study was to explore the relationships of GLS with TDI measurements of s' and SExc, LVEDL and heart rate. The study was performed in selected subjects with a LVEF within the normal range and no ischemic or valvular heart disease, but a low e', and thus increased risk of cardiovascular disease. These criteria resulted in a group in which GLS varied over a range from normal to moderately reduced. The main findings were: (1) the absolute value of GLS was positively correlated with both s' and SExc, but was more closely correlated with SExc, (2) although not significant on univariate analysis, the absolute value of GLS was inversely correlated with LVEDL in multivariate models where LVEDL was combined with SExc, and (3) the prediction of GLS was improved by the inclusion of heart rate in models which included s', and to a lesser extent, in models which included SExc.

**Table 5. Reproducibility measurements.**

| | Observer 1 | Observer 2 | r | Difference (SD) |
|---|---|---|---|---|
| Absolute GLS (%) | 17.4±2.5 | 17.6±2.6 | 0.71 | 0.2 (1.9) |
| s'(cm/s) | 6.6±1.2 | 6.8±1.2 | 0.93 | 0.2 (0.4) |
| SExc (cm) | 1.1±0.2 | 1.2±0.2 | 0.89 | 0.1 (0.1) |

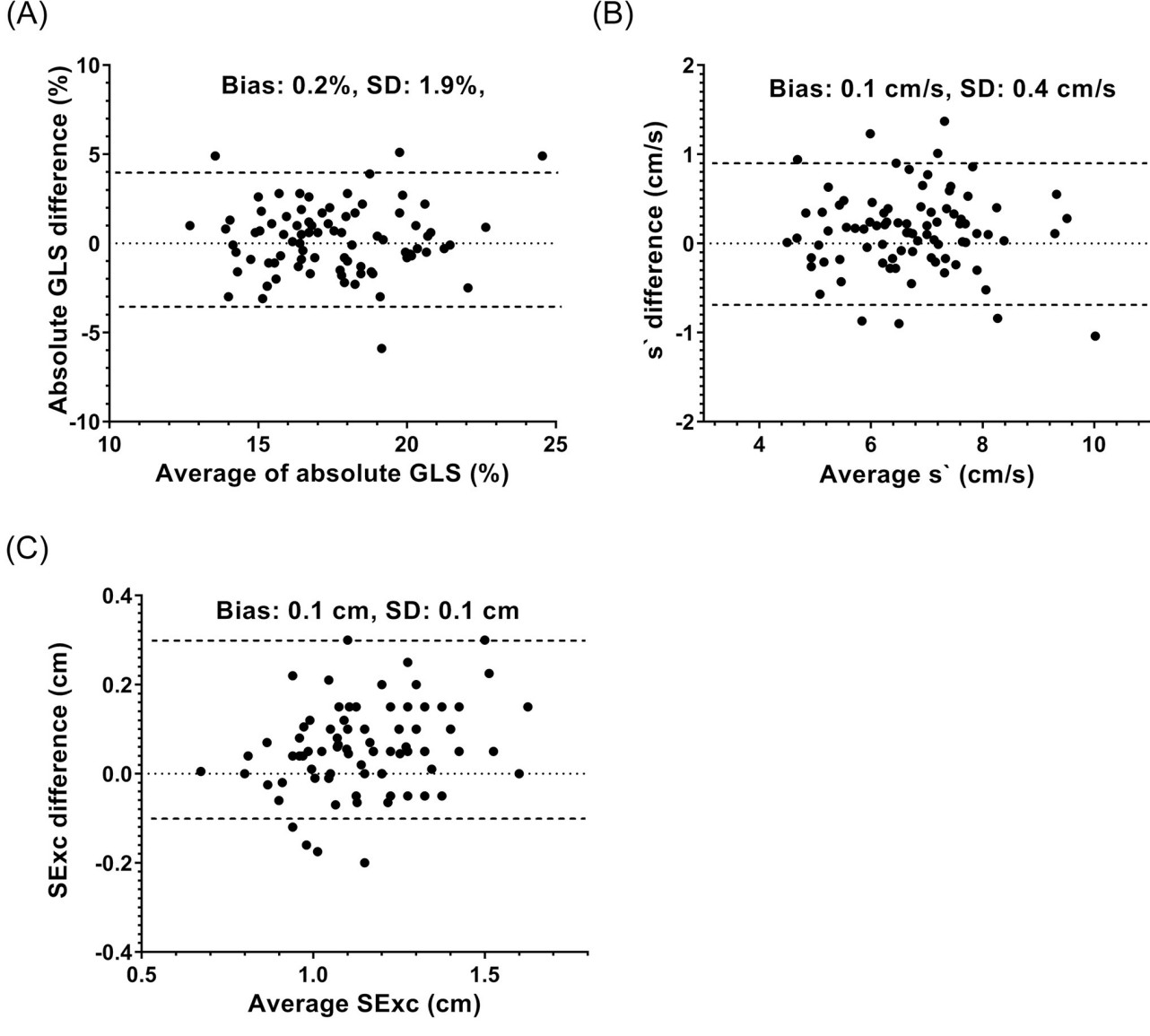

**Fig 3. Bland-Altman plots of the results from the two observers for GLS (A), s' (B) and SExc (C).** On each plot there is a horizontal dotted line at y = zero representing the position of no difference between the two observers, and two horizontal dashed lines representing the 95% confidence limits of the differences between the two observers.

There is evidence that GLS has prognostic significance in subjects with a preserved LVEF [11, 16, 17], and that GLS is a more sensitive detector of anthracycline cardiotoxicity than LVEF [28]. There is also evidence for the prognostic significance of the relatively simpler measurements of s' [14, 22, 23, 29, 30] and SExc [12, 13] in subjects with a preserved LVEF, and reports of greater sensitivity of s' compared with LVEF for detection of early anthracycline toxicity [31]. However, whether GLS, s' and SExc provide similar, and thus possibly interchangeable information with respect to the assessment of LV long-axis systolic function has received little attention in previous studies. The presence of correlations of GLS with s' and SExc in our study are consistent with the findings of previous studies [3, 9, 11, 32], but previous studies did not compare the relationships of GLS with s' and SExc within the one study. The observation of a closer relationship between GLS and SExc compared to that between GLS and s' may have

practical importance given that assessment of LV long axis function by TDI has been most commonly performed using peak velocities. In conjunction with previous evidence for the prognostic significance of SExc, the findings of the present study not only provide support for considering SExc in future attempts to better understand the nature and significance of GLS, but also for its inclusion in studies in which monitoring is being performed for a deterioration in LV long-axis contraction and where GLS may not be feasible.

While there is evidence that s' and SExc both provide information about LV long-axis function [4, 31, 33], there has been little investigation of the relationship of s' with SExc (as measured by any of the available techniques of 2D, M-mode or TDI integrals) in the same study. In the present study, heart rate had an important modifying effect on the relationship between s' and SExc. Although not a correlate of s' on univariate analysis, heart rate became a positive correlate and independent contributor to a model of s' when combined with SExc. A similar finding has been reported in other groups of subjects with a normal LVEF [21], and can be most simply explained by consideration of the duration of the systolic signal, which becomes shorter as the heart rate increases [21]. Thus, if SExc remains the same at a higher heart rate but with a shorter contraction duration, then the peak velocity (s') would be expected to increase. Nevertheless, this simple explanation may be incomplete as it does not take into account a number of potentially important factors such as the effects of the force frequency relationship on the extent of contraction, reductions in heart rate which can occur after endurance training and with subject fitness, and that an increase in heart rate could be a compensation for diminished contraction [21]. Indeed, highlighting the complexity of heart rate relationships in the present study was that heart rate not only modified the relationship between s' and SExc, but was also an inverse correlate of SExc.

Heart rate also had an important modifying effect on the relationships of GLS with SExc and s' in the present study. Thus, heart rate was a significant and substantial contributor in models of GLS when combined with s', with this inverse correlation of heart rate with absolute GLS at least partly explained by it acting as a correction factor for the relationship of s' and SExc in the setting of GLS being more closely related to SExc than s'. An inverse correlation of absolute GLS with heart rate has been previously reported in HFpEF [9] and healthy subjects [34], however, no explanation for these observations has been proposed. That heart rate was also a contributor to the prediction of GLS when combined with SExc indicates that the mechanism is more complex than just an adjustment for the relationship between s' and SExc. The inverse correlations of the absolute value of GLS and SExc with heart rate are consistent with the possibility that heart rate may increase as a compensation for diminished long-axis contraction. Consideration of heart rate could be important when changes in s', SExc or GLS are observed in individual subjects during follow-up studies as it is likely that the heart rates of individuals will vary between studies. An effect of heart rate could even become a systematic issue with respect to the assessment of long-axis function in some circumstances, e.g. heart rate has been reported to increase after anthracycline treatment [35–37].

LVEDL was an independent inverse correlate of the absolute value of GLS when combined with SExc in the present study and an interaction between LVEDL and SExc was also evident, as the partial correlation coefficients of SExc and LVEDL both increased in the model of GLS when they were combined. This finding was not unexpected as GLS reflects percentage deformation and should have a relationship with long-axis fractional shortening, which is also expressed as a percentage, the latter variable related to both the extent of systolic excursion and the end-diastolic length. Although not directly addressed by the current study, our finding invites speculation of how the variability of LVEDL between individuals and LVEDL change over time might modify the relationship between SExc and GLS in different clinical conditions. Thus, shortening of LVEDL has been reported to occur in association with aging [26, 38,

39], whereas an increase in LVEDL might be expected in association with LV dilatation, such as is seen during medium-term follow-up of anthracycline toxicity [40]. On the other hand, there have now been several reports that LV end-diastolic volume can decrease within 3 months of the commencement of chemotherapy [41], although information about what happens to LVEDL in these circumstances is not available. Nevertheless, that LVEDL has the potential to increase and decrease in individuals over time could be important when interpreting changes in s', SExc and GLS in future longitudinal studies.

In previous studies, GLS has not been feasible in a substantial proportion of the subjects and this is one of the reasons why alternative methods for the assessment of long-axis LV function such as TDI systolic variables merit consideration. In the HUNT study of a healthy population of 1266 individuals only 13765 of 22788 (60%) of analyzed segments yielded optimal speckle tracking and in randomized trials in HFpEF, inadequate imaging quality for strain measurement was reported in 56/477 subjects by Shah et al [11], and in 82/301 subjects by Kraigher-Krainer et al [9]. Furthermore, in one population study, 858/2154 subjects were excluded from strain measurement because of inadequate frame rate or inadequate image quality, and in the remaining 1296 subjects strain could be measured in all 3 apical projections in only 284 subjects [17]. In our study group speckle strain measurement was possible in all 3 apical projections in nearly all (84 out of 87) of the subjects, but this feasibility is not directly comparable to the above studies as subjects in our study were excluded from inclusion if the 2D image quality in any of the 3 views did not appear suitable for GLS measurement. Consistent with the high reported feasibility of TDI measurements in population studies [19], TDI measurements could be made in all subjects in the present study.

We found close correlations between the two observers for both of the TDI variables, but only a moderate correlation between the two observers for GLS. The presence of a moderate correlation for GLS is consistent with the finding of a large study in subjects with HFpEF in which the interobserver correlation coefficient was identical to that of the present study at 0.71 [11]. On the other hand, there were small systematic differences in both of the TDI variables, as well as GLS between the 2 observers. Although there is a rationale why SExc obtained from the 4 chamber view may be more closely correlated with 4 chamber GLS than with the standard average of 3 views GLS, the opposite was found in our study. One possibility for this finding, in a group of subjects in which genuine differences in GLS from the different views were not expected, is that averaging of GLS from the 3 views may provide a partial correction of random errors in the 4 chamber view results and thus a GLS result closer to the true value.

This study is an observational study and thus there are inherent limitations related to the conclusions which can be drawn. We specifically studied a group with low e'because this group is at higher risk of cardiac events, and also because moderate variation in the magnitude of GLS was both expected and necessary to allow assessment of the relationship of GLS with TDI variables. It cannot be assumed that the relationships we found would be present in subjects selected on the basis of different criteria. Also, an unavoidable limitation of this study, and indeed of any study comparing GLS with other variables, is that it is not possible to assess GLS relationships in subjects in which GLS cannot be measured. There would have been variations in the magnitude of all the studied variables due to minute to minute natural variation, variations in acquisition and imprecision in measurement. Therefore the magnitude of the correlations between the different variables may well be an underestimate of the actual relationships had all signals and apical loops been able to be obtained simultaneously and optimally, and measured without error [42].

In conclusion, GLS is notionally similar to long-axis LV fractional shortening and consistent with this, in the present study the absolute value of GLS was positively correlated with SExc and inversely correlated with LVEDL. GLS was more closely related to SExc than s', at

least partly related to a modifying effect of heart rate on the relation between SExc and s'. Given that the available evidence suggests that mitral annular TDI measurements are possible in most subjects, whereas GLS is not obtainable in a substantial proportion of subjects, our findings provide a rationale for the measurement of SExc from the septal and lateral LV walls in future studies where the aim is for the sensitive detection of a deterioration in long-axis function.

## Supporting information

**S1 File. GLS relationships study.**
(XLSX)

## Author Contributions

**Conceptualization:** Roger E. Peverill, Kevin Cheng, James Cameron.

**Data curation:** Roger E. Peverill, Kevin Cheng.

**Formal analysis:** Roger E. Peverill, Kevin Cheng.

**Investigation:** Roger E. Peverill, Kevin Cheng, Lesley Donelan.

**Methodology:** Roger E. Peverill, Kevin Cheng, Lesley Donelan.

**Project administration:** Roger E. Peverill, Kevin Cheng, Lesley Donelan.

**Resources:** Roger E. Peverill.

**Software:** Roger E. Peverill, Kevin Cheng.

**Supervision:** Roger E. Peverill.

**Validation:** Roger E. Peverill.

**Writing – original draft:** Roger E. Peverill.

**Writing – review & editing:** Roger E. Peverill, Kevin Cheng, James Cameron, Lesley Donelan.

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
