## [Decision Letter · Decision Letter 0]

13 May 2020

PONE-D-20-06704

Relationships of global longitudinal strain with s`, long-axis systolic excursion, left ventricular length and heart rate

PLOS ONE

Dear Dr Peverill,

Thank you for submitting your manuscript to PLOS ONE. After careful consideration, we feel that it has merit but does not fully meet PLOS ONE’s publication criteria as it currently stands. Therefore, we invite you to submit a revised version of the manuscript that addresses the points raised during the review process.

The authors investigates relation of GLS with TDI measures. The comments of the reviewer were conflicting. In my opinion findings are not innovative, moreover there are methodological concerns especially related to the retrospective nature of the study.  We will reconsider the paper after a comprehensive revision. 

We would appreciate receiving your revised manuscript by Jun 27 2020 11:59PM. To enhance the reproducibility of your results, we recommend that if applicable you deposit your laboratory protocols in protocols.io, where a protocol can be assigned its own identifier (DOI) such that it can be cited independently in the future. For instructions see: http://journals.plos.org/plosone/s/submission-guidelines#loc-laboratory-protocols

We look forward to receiving your revised manuscript.

Kind regards,

Giuseppina Novo

Academic Editor

PLOS ONE

Journal Requirements:

2. In your Methods section, please provide additional information about the participants included in the analysis. Please ensure you have provided sufficient details to replicate the analyses such as: a) a description of any inclusion/exclusion criteria that were applied to participant inclusios, c) a table of relevant demographic details, d) a statement as to whether your sample can be considered representative of a larger population, e)  descriptions of where participants were recruited and where the research took place.

Reviewers' comments:

Reviewer's Responses to Questions

**Comments to the Author**

1. Is the manuscript technically sound, and do the data support the conclusions?

Reviewer #1: Yes

Reviewer #2: Partly

Reviewer #3: Yes

2. Has the statistical analysis been performed appropriately and rigorously? 

Reviewer #1: Yes

Reviewer #2: No

Reviewer #3: Yes

3. Have the authors made all data underlying the findings in their manuscript fully available?

Reviewer #1: Yes

Reviewer #2: No

Reviewer #3: Yes

4. Is the manuscript presented in an intelligible fashion and written in standard English?

Reviewer #1: Yes

Reviewer #2: Yes

Reviewer #3: Yes

5. Review Comments to the Author

Reviewer #1: Your manuscript is interesting and well written. In your manuscript you investigated the relationships of GLS with TDI measurements of SExc and s`, LVEDL and heart rate in a group of subjects without a history of heart failure.Your manuscript is interesting in the purpose and results. It is technically sound and well written.

Reviewer #2: The authors investigates relation of GLS with TDI measures. They concludes that GLS is more closely related to Long-axis systolic excursion (SExc) than peak systolic velocity (s’) of mitral annular motion, and is also independently and inversely related to LV end-diastolic length (LVEDL) and heart rate.

It is well known that SExc and s’ are additional technique to evaluate systolic function even if ejection fraction is within normal values. Therefore, this observation is not innovative.

Several studies have shown that GLS is reduced in particular setting of patient with normal EF and this alteration has important therapeutic implications. Techniques to evaluate systolic function proposed by authors can be used when the images are not of good quality. In my opinion manuscript is not innovative.

This is a retrospective study and analysis was performed on stress echocardiograms and this is a strong limit. GLS evaluated in a retrospective study on stress echocardiograms is not reliable and probably this justifies the results of the study.

For these reasons, in my opinion manuscript is not suitable for publication.

Reviewer #3: The authors demonstrated, in subjects with normal LVEF but reduced e`, that the absolute value of GLS is more closely related to SExc than s`, and that measurement of SExc may provide an additional or alternative technique for the assessment of LV long-axis function when GLS is not available or not feasible.

Some minor comments:

- The study claims are convincing and well supported by the experimental data.

- The manuscript is clearly written and fair in the treatment of previous literature. However, introduction and discussion sections in some parts are too long and difficult to follow.

- Methods: methodology of SExc measurements needs to be described.

- The images are coherent, however not easy to interpret. Please provide descriptive figure legends.

- Please add the reference lines (+/- 1,96 SD) to the Bland Altman plots;

- A figure with scatter plots describing the GLS linear correlations may render the paper more accessible.

6. PLOS authors have the option to publish the peer review history of their article (what does this mean?). If published, this will include your full peer review and any attached files.

Reviewer #1: No

Reviewer #2: No

Reviewer #3: No

---

## [Author Response · Author response to Decision Letter 0]

23 May 2020

Author response: The manuscript has been checked that it meets PLOS ONE requirements.

2. In your Methods section, please provide additional information about the participants included in the analysis. Please ensure you have provided sufficient details to replicate the analyses such as: 

a) a description of any inclusion/exclusion criteria that were applied to participant inclusions,

Author response: A number of exclusion criteria were already listed in the Subjects subsection of the Methods section but an adjustment has also been to this subsection to provide some additional information about the exclusion and inclusion criteria for the study. 

 c) a table of relevant demographic details, 

Author response: Table 1 already provides relevant demographic details about the subjects.

d) a statement as to whether your sample can be considered representative of a larger population, 

Author response: The main aim of this study was methodological, with the subject selection largely based on methodological issues and not on the desire to describe a particular population. There is already a limitations paragraph in the Discussion which says that It cannot be assumed that the relationships we found would be present in subjects selected on the basis of different criteria. A statement has been added to this paragraph related to the methodological issue (which applies to any study in which the relationship of GLS with other variables is examined) that if GLS is not measurable in a subject then it cannot be compared to TDI variables.

e) descriptions of where participants were recruited and where the research took place.

Author response: There is now an additional phrase in the Subjects subsection which indicates that the echocardiographic studies were performed in our institution (Monash Health).

Author response: The affiliation for each author has been added.

Comments to the Author

We thank all the reviewers for their interest and comments.

Reviewer #1: Your manuscript is interesting and well written. In your manuscript you investigated the relationships of GLS with TDI measurements of SExc and s`, LVEDL and heart rate in a group of subjects without a history of heart failure. Your manuscript is interesting in the purpose and results. It is technically sound and well written.

Author response: We are very pleased that Reviewer 1 found the manuscript to be interesting, technically sound and well written. 

Reviewer #2: The authors investigates relation of GLS with TDI measures. They concludes that GLS is more closely related to Long-axis systolic excursion (SExc) than peak systolic velocity (s’) of mitral annular motion, and is also independently and inversely related to LV end-diastolic length (LVEDL) and heart rate.

It is well known that SExc and s’ are additional technique to evaluate systolic function even if ejection fraction is within normal values. Therefore, this observation is not innovative.

Several studies have shown that GLS is reduced in particular setting of patient with normal EF and this alteration has important therapeutic implications. Techniques to evaluate systolic function proposed by authors can be used when the images are not of good quality. In my opinion manuscript is not innovative.

Author response: Reviewer 2 seems to have misunderstood the aim of this study. There was no suggestion made by the authors that the techniques of s` and SExc were considered innovative. Indeed, that s`, SExc and GLS had all been studied before in subjects with a normal EF was clearly described in the Introduction and elaborated further in the Discussion. However, whether the systolic TDI variables provide the same information as GLS cannot be assumed. The aim of this study was to investigate the relationships of GLS with s`, SExc, LVEDL and heart rate, a systematic investigation of which is new, as to the best of our knowledge it has not been performed previously. That GLS is more closely correlated with SExc and s` has not been previously reported. 

This is a retrospective study and analysis was performed on stress echocardiograms and this is a strong limit. GLS evaluated in a retrospective study on stress echocardiograms is not reliable and probably this justifies the results of the study.

For these reasons, in my opinion manuscript is not suitable for publication.

Author response: The authors can see no implications for the conclusions of the study related to the subjects being identified retrospectively, particularly as the exclusion and inclusion criteria are clearly stated in the manuscript. Neither can the authors understand why there might be an objection that the study group had all undergone stress echocardiography as stress echocardiography has been performed in previous studies in which inducible ischaemia was an exclusion criteria for the study. We also have tried to address the possibility that there was a misunderstanding regarding the timing of the imaging, with Reviewer 2 possibly thinking that GLS and TDI was performed on images obtained after exercise. GLS, s` and SExc were measured in all subjects on baseline (resting) images obtained with the subjects lying in a left lateral position. The Methods section has been changed to make this clear.

Reviewer #3: The authors demonstrated, in subjects with normal LVEF but reduced e`, that the absolute value of GLS is more closely related to SExc than s`, and that measurement of SExc may provide an additional or alternative technique for the assessment of LV long-axis function when GLS is not available or not feasible.

Some minor comments:

- The study claims are convincing and well supported by the experimental data.

- The manuscript is clearly written and fair in the treatment of previous literature. However, introduction and discussion sections in some parts are too long and difficult to follow.

Author response: We are very pleased that Reviewer 3 found the study claims convincing and well supported by the experimental data and the manuscript to be clearly written. Without any specific sections highlighted as being problematic, we have not been able to address this issue raised by the Reviewer that some parts of the manuscript being too long or difficult to follow.

- Methods: methodology of SExc measurements needs to be described.

Author response: An addition has been added to the Methods section with regard to SExc measurement.

- The images are coherent, however not easy to interpret. Please provide descriptive figure legends.

Author response: A more detailed figure legend has now been included for the Bland Altman plots (now Fig 3A, 3B & 3C in the revised manuscript). Because we did not want the way the data was presented to imply any significance from there being different signs for the biases of GLS and the TDI variables, Table 5 showing reproducibility measurements has been altered so that the one observer who measured both GLS and TDI variables is now shown as Observer 1 and the two other observers, one who measured GLS and one who measured TDI variables, are now shown as Observer 2. This results in all the biases being positive and the plots have been adjusted to show this. The alterations do not change anything fundamental about the interpretation regarding reproducibility.

- Please add the reference lines (+/- 1,96 SD) to the Bland Altman plots;

Author response: The lines corresponding to the 95% confidence limits (+/- 1,96 SD) have been added to the Bland Altman plots

- A figure with scatter plots describing the GLS linear correlations may render the paper more accessible.

Author response: There are now figures included which show scatter plots and the correlations of GLS with s` and SExc (Fig 1 and Fig 2).

---

## [Decision Letter · Decision Letter 1]

15 Jun 2020

PONE-D-20-06704R1

Relationships of global longitudinal strain with s`, long-axis systolic excursion, left ventricular length and heart rate

PLOS ONE

Dear Dr. Peverill,

Thank you for submitting your manuscript to PLOS ONE. After careful consideration, we feel that the manuscript is improved  but does not fully meet PLOS ONE’s publication criteria as it currently stands. Therefore, we invite you to submit a revised version of the manuscript that addresses the points raised during the review process.

We look forward to receiving your revised manuscript.

Kind regards,

Giuseppina Novo

Academic Editor

PLOS ONE

Reviewers' comments:

Reviewer's Responses to Questions

**Comments to the Author**

1. If the authors have adequately addressed your comments raised in a previous round of review and you feel that this manuscript is now acceptable for publication, you may indicate that here to bypass the “Comments to the Author” section, enter your conflict of interest statement in the “Confidential to Editor” section, and submit your "Accept" recommendation.

Reviewer #1: All comments have been addressed

Reviewer #3: All comments have been addressed

2. Is the manuscript technically sound, and do the data support the conclusions?

Reviewer #1: (No Response)

Reviewer #3: Yes

3. Has the statistical analysis been performed appropriately and rigorously? 

Reviewer #1: (No Response)

Reviewer #3: Yes

4. Have the authors made all data underlying the findings in their manuscript fully available?

Reviewer #1: (No Response)

Reviewer #3: Yes

5. Is the manuscript presented in an intelligible fashion and written in standard English?

Reviewer #1: (No Response)

Reviewer #3: Yes

6. Review Comments to the Author

Reviewer #1: (No Response)

Reviewer #3: The manuscript has been improved considerably. I am happy with the edits that were made in response to my earlier review.

7. PLOS authors have the option to publish the peer review history of their article (what does this mean?). If published, this will include your full peer review and any attached files.

Reviewer #1: No

Reviewer #3: No

---

## [Author Response · Author response to Decision Letter 1]

19 Jun 2020

I understand from the Editor that the revised manuscript was accepted and that no further changes are required.

Roger Peverill

---

## [Editor Report · Decision Letter 2]

23 Jun 2020

Relationships of global longitudinal strain with s`, long-axis systolic excursion, left ventricular length and heart rate

PONE-D-20-06704R2

Dear Dr. Peverill,

We’re pleased to inform you that your manuscript has been judged scientifically suitable for publication and will be formally accepted for publication once it meets all outstanding technical requirements.

Kind regards,

Giuseppina Novo

Academic Editor

PLOS ONE

Additional Editor Comments (optional):

Dear Author,

I apologize for the inconvenience.
---

## [Editor Report · Acceptance letter]

10 Jul 2020

PONE-D-20-06704R2 

Relationships of global longitudinal strain with s`, long-axis systolic excursion, left ventricular length and heart rate 

Dear Dr. Peverill:

I'm pleased to inform you that your manuscript has been deemed suitable for publication in PLOS ONE. Congratulations! Your manuscript is now with our production department. 

Kind regards, 

on behalf of

Professor Giuseppina Novo 

Academic Editor

PLOS ONE